# Self-Supervised Visual Preference Alignment

## ABSTRACT

This paper makes the first attempt towards unsupervised preference alignment in Vision-Language Models (VLMs). We generate chosen and rejected responses with regard to the original and augmented image pairs, and conduct preference alignment with direct preference optimization. It is based on a core idea: properly designed augmentation to the image input will induce VLM to generate false but hard negative responses, which helps the model to learn from and produce more robust and powerful answers. The whole pipeline no longer hinges on supervision from GPT-4 or human involvement during alignment, and is highly efficient with few lines of code. With only 8k randomly sampled unsupervised data, it achieves 90% relative score to GPT-4 on complex reasoning in LLaVA-Bench, and improves LLaVA-7B/13B by 6.7%/5.6% score on complex multi-modal benchmark MM-Vet. Visualizations shows its improved ability to align with user-intentions. A series of ablations are firmly conducted to reveal the latent mechanism of the approach, which also indicates its potential towards further scaling.

## CCS CONCEPTS

• **Computing methodologies** → **Computer vision**; *Natural language generation*.

## KEYWORDS

Vision-Language Models, Self-Supervised, Direct Preference Optimization

## 1 INTRODUCTION

Large Vision-Language Models have recently emerged and greatly advanced current researches [2, 27, 36, 41]. Representative methods in this family, such as MiniGPT4 [47], LLaVA [27] and Instruct-BLIP [8], try to properly align vision signals to Large Language Models (LLMs) to better conduct multi-modal comprehension. These methods usually undergo a pretraining stage with abundant image-text pairs for modality alignment before supervised finetuned (SFT) with academic [8] or GPT-4 generated [4, 43] SFT data. There are variants of them (e.g., LLaVA-based) that try to improve the VLMs' instruction following ability by leveraging high quality pretraining pairs [4, 24] or scaling up SFT database [23, 38, 43].

Despite their success [4] in boosting the comprehension skills of current VLMs, these models are not properly aligned with user-intentions. As a result, they lack the ability to reject samples and could induce more unintended output. For instance, Instruct4V [38]

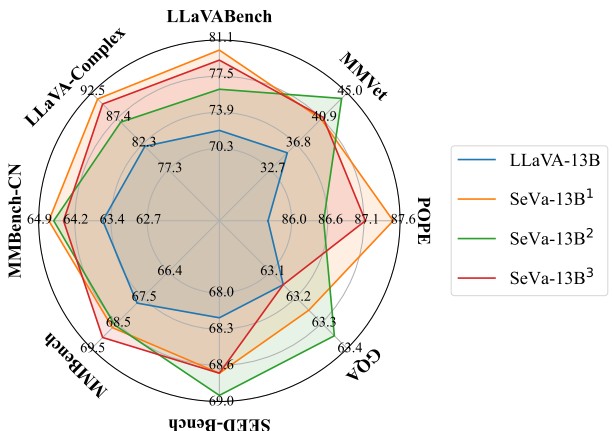

**Figure 1: Illustration of the baseline LLaVA-13B (its 1.5 version) and the proposed SeVa-13B paradigm. Here we demonstrate three variants of SeVa with different sampled seed to obtain the un-labeled dataset (the image-text pairs used for DPO sample generation, cf. Alg. 1).**

construct 220k SFT data from GPT-4, but still struggle on hallucination benchmark POPE [21]. The same can be observed in Fig. 7, where LLaVA failed to follow user instructions and provide meaningless information. Recently, there *are* few trials that incorporate alignment techniques into vision-language fields [20, 37, 44]. However, we found that they only emphasize on quite limited task domains [44], and, more importantly, *their data construction pipeline requires extra knowledge source either from GPT-4 or human feedbacks*. This might hinder them from further data scaling, as preference data is *not cheap* (e.g., a 10k collected human-evaluated instances requires a cost of 3000\$ [22]).

In this paper, we propose a self-supervised paradigm that can *freely* obtain arbitrary number of preference samples with minimal code implementation (cf. Alg. 1). Simply but, we have enjoyed the advantage of this pipeline and observed a significant improvement in the capabilities of current VLMs from various aspects: stronger chain-of-thought skills, better OCR [30] ability, proper alignment with user-intentions, less hallucinations, etc (cf. Fig. 7).

Our motivation came from an interesting discovery in Fig. 2, where we apply commonly adopted image augmentations in *visual contrastive learning* [13, 48, 49] during LLaVA inference. The effect of all these test-time augmentations are evaluated on three common multi-modal benchmarks (MMVet [40], MMBench [28], POPE [21]). It is not surprising to observe in Fig. 2 that *vision-language models are quite sensitive to data-augmentations* and, slightly distortion will induce the model to output different semantic tokens. Then, an natural question arises: *are the original and distorted responses valuable to construct preference data for DPO training?* Since this will totally free the data construction from *any* extra source, and might relieve the difficulty of scaling up preference data as well [42].

**Algorithm 1** Pseudocode of SeVa in a PyTorch style.

```
# Q, I: question set, image set
# model: the SFT vision-language model
# C, R: chosen and rejected answer set
# T: sampled data augmentation

# randomly sample data pair, generate answer
Q, I = random.sample(data)
C, R = model(I, Q), model(T(I), Q)

# filtering equal answers
idx = (C != R)
C_t, R_t = C[idx], R[idx]
Q_t, I_t = Q[idx], I[idx]

# DPO training, omit reference model for simplicity
DPOTrain(model, (Q_t, I_t), (C_t, R_t))
```

Our motivation was then firmly verified by quantitative experiments in Table 1, in which we randomly sampled 8k image-question pairs from the subset of LLaVA665k [26] (cf. Sec. 4.1 for details), and choose 5 augmentations to generate the preference data using LLaVA-7B. All these self-generated data are again fed into the *same* LLaVA-7B model for DPO training. As Table 1 shows, all the augmentations are helpful for model comprehension, especially on GPT-4 evaluated benchmark MM-Vet, which makes our motivation valid. In the meanwhile, we found that either too strong (e.g., Diffusion-S) or too weak data augmentation (e.g., RandFlip) are sub-optimal for the whole pipeline, and medium is the best (e.g., Diffusion-W, MOCO). Our final conjecture is: *self-generated augmentation pairs are suitable preference sample to improve multi-modal abilities, and hard negatives are most preferred.*

We name our methodology as: **Se**lf-supervised **V**isual preference **a**lignment (SeVa), and summarize its whole pipeline in Alg. 1. Though being embarassingly simple with few lines of code to implement, we mathematically showcase its strong relation with visual contrastive learning in maxmizing a preference distribution, where SeVa could probably be viewed as a special form of contrastive learning with one negative sample. This makes SeVa easily extendable if more negatives are involved (cf. appendix).

Experiments in common multi-modal benchmarks demonstrate the effectiveness of our method, where SeVa improves the VLM's comprehension ability by a large margin (e.g., SeVa-7B even surpasses LLaVA-1.5-13B by 1.8% in MMVet). In addition, we carefully design detailed ablations to reveal the latent mechanism of SeVa from different angles. We found that SeVa shows surprisingly results like detailed descriptions, less hallucinations, stronger OCR skills and chain-of-thought ability, etc. Our contributions are:

- For the first time, we conduct visual preference alignment in an un-supervised manner. The whole pipeline *do not* require any GPT-4 or costly human annotated data.
- We found such an alignment pipeline have numerous benefits such as enhanced multi-modal comprehension ability, better alignment with user-intentions, etc. Visualizations in Fig. 7 and ablations firmly verify our hypothesis as well.
- Our method, SeVa, enjoys efficiency in pipeline and simplicity in implementation, which paves way for future preference alignment in visual-language domain.

## 2 RELATED WORKS

**Large Vision-Language Models (VLMs).** The great development of Large Language Models (LLMs) has facilitated the community in both academics [2, 8, 19, 26, 47] and industries [1, 7, 36, 46]. Recently, large vision-language models (VLMs) emerged, aiming to extend the reasoning brain of LLM to vision modality. The majority of VLMs undergo a two-stage training manner, with the pretraining period focusing on aligning the vision and text signals before finetuned with instruction following data in the second stage. LLaVA [27], as one of its representatives, has attracted numerous researchers [4, 22, 23, 38, 39] since it provide a valuable opportunity for them to reproduce and built upon. Subsequent works based on LLaVA try to leverage more image-text data pairs of higher quality [4] or to construct more abundant instruction following database [38]. Despite their achievements, these model are not preference aligned with user intentions neither implicitly nor explicitly, which might limit their further deployment. Therefore, an suitable alignment technique is of great importance.

**Preference alignment in LLM/VLM.** Training an LLM to align with human or user preference is called RLHF [31], which usually happens after the instruction following stage. The core concept of RLHF is to reduce un-intented or toxic output produced by LLMs [10, 50]. Variants of RLHF include, but not limited to, DPO [33], PPO [34] and RLAIF [3]. Alignment has been fruitfully researched in natural language processing (NLP) domains [3, 12], but relatively less visited in vision-language fields. There are some latest attempts [20, 37, 44] that integrate alignment techniques in vision-language domains. However, they only emphasize on task specific ability (e.g. hallucinations in HA-DPO [44]), and, more importantly, it requires tedious GPT-4 or human interventions to construct the preference data, which could diminish the pipeline efficiency. In this paper, we propose an unsupervised data construction pipeline that not only solve the data hungry issue in preference alignment, but also greatly boosts the comprehension ability of current VLMs, as well.

**Contrastive learning.** Self-supervised learning (SSL) of visual representation are popularized in the past years [5, 11, 13, 48]. Traditional SSL are mostly contrastive based, where strongly augmented positive views of the same image *will* share similar deep semantic in the hidden space that the model could capture. Nevertheless, we found that a similar augmentation pipeline *does not* holds true in VLMs. Inspired by unsupervised property of SSL, we design an self-supervised pipeline to construct the preference data, and empower the current VLM with improved capability.

## 3 METHOD

We will first show the preliminaries of language modeling and direct preference optimization, then move onto the procedure of the proposed SeVa pipeline. Finally, we discuss its strong relation with visual contrastive learning.

### 3.1 Preliminaries

**Language modeling.** We start from the vison-language modeling perspective and take LLaVA [27] as an example for illustration. Given an image input $I$, it is first passed through a vision encoder

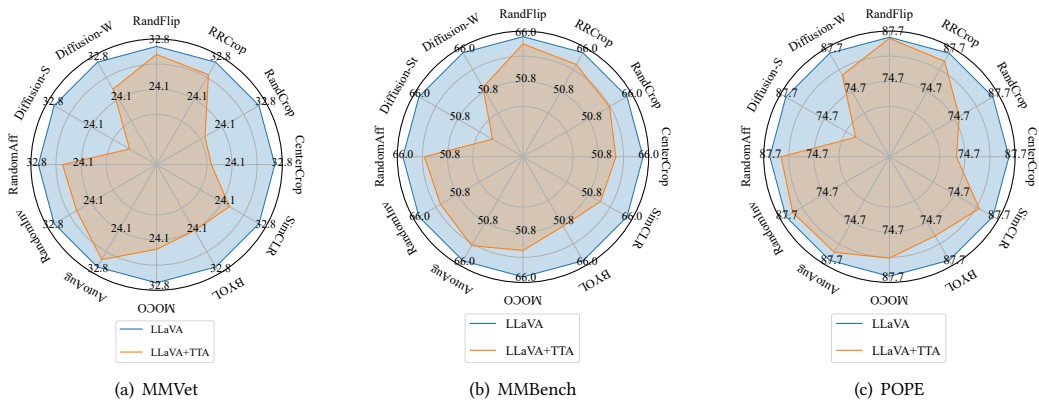

Figure 2: The test-time image augmentation (TTA) plugged into LLaVA-1.5 on three benchmarks. We involve standard strategies: RandFlip, RandomResizedCrop ('RRCrop'), RandCrop, CenterCrop, RandomAffine, RandomInvert and AutoAugmentation; diffusion noise: Diffusion-Weak ('W') and Diffusion-Strong ('S'); mixtures: augmentations adopted in MOCO, BYOL and SimCLR.

Table 1: Six data augmentations selected to generate preference data for DPO training (w/ LLaVA-1.5). Each strategy was adopted to distort the image (together with the questions) to produce rejected response, which is paired with the chosen response from the original image input (cf. Alg. 1). The models after DPO training with these preference pairs are then evaluated on MMVet [40], MMBench [28] (in english 'en' and chinese 'cn') and POPE [21] benchmarks, respectively (cf. Sec. 3.2 for details).

| DPO data | MMVet | | | | | | | MMBench | | POPE | | | |
| | All | rec | ocr | know | gen | spat | math | en | cn | All | rand | pop | adv |
| --- | --- | --- | --- | --- | --- | --- | --- | --- | --- | --- | --- | --- | --- |
| N/A | 30.5 | 35.7 | 21.9 | 17.4 | 19.7 | 24.7 | 7.7 | 64.3 | 58.3 | 85.9 | 89.5 | 86.7 | 81.7 |
| RandFlip | 33.7 | 37.2 | 26.4 | 21.8 | 23.9 | 29.1 | 7.7 | 64.4 | 58.3 | 86.7 | 89.2 | 87.1 | 83.9 |
| RRCrop | 33.8 | 39.2 | 23.2 | 21.9 | 24.5 | 27.7 | 7.7 | 65.5 | 59.5 | 86.8 | 89.4 | 87.4 | 83.6 |
| AutoAug | 35.0 | 38.5 | 26.8 | 22.9 | 24.4 | 29.3 | 15.0 | 65.4 | 59.6 | 86.5 | 88.8 | 86.8 | 83.9 |
| Diffusion-W | 37.2 | 40.2 | 29.9 | 21.8 | 23.9 | 34.3 | 11.2 | 65.6 | 59.2 | 86.7 | 89.4 | 87.1 | 83.6 |
| MOCO | 37.0 | 40.9 | 29.0 | 21.3 | 25.6 | 32.5 | 15.0 | 65.2 | 59.8 | 86.6 | 89.1 | 87.1 | 83.7 |
| Diffusion-S | 34.6 | 38.8 | 26.5 | 20.5 | 23.4 | 32.0 | 11.5 | 65.2 | 58.2 | 86.6 | 89.2 | 87.5 | 83.3 |

(e.g., ViT-L [32]) to get:

$$\boldsymbol{h} = g(I), \qquad (1)$$

where $g$ denotes a combination of vision encoder and projection layer. $\boldsymbol{h}$ represents a sequence of deep image embeddings. The embeddings before the last transformer layer are considered in LLaVA-1.5 [26], which are then concatenated with the question token $q$ and fed into the auto-regressive LLM $\pi$ that sequentially generates the next target token $y$:

$$\pi_\theta(y|x) = \prod_{i=1}^{L} \pi_\theta(y_i|y_{<i}, x), \qquad (2)$$

where $x = (\boldsymbol{h}, q)$, and is considered as prompt sent to the LLM parameterized by $\theta$. $y_{<i}$ is the generated token before the current prediction $y_i$, and $L$ is the length of the produced token sequece. We denote the LLM trained with SFT data as $\pi_{\text{SFT}}$ for clarity.

**Direct preference optimization.** DPO is first proposed in [33], and can be viewed a new parameterization of the reward model in RLHF [31] that can directly optimize the policy (the LLM's parameter $\theta$). Here we start with RLHF for better illustrations. In RLHF, a Bradley-Terry (BT) reward model is often adopted, which stipulates

the human preference distribution as:

$$p^*(y_c > y_r|x) = \frac{\exp\left(r^*(x, y_c)\right)}{\exp\left(r^*(x, y_c)\right) + \exp\left(r^*(x, y_r)\right)}. \qquad (3)$$

The $y_c$ and $y_r$ are the chosen and rejected response conditioned on the LLM's prompt $x$. The reward model $r_\phi(\cdot)$ is paramerized by $\phi$ and can be obtained with maximum likelihood using preference database $\mathcal{D}$ annotated by human [33]:

$$\mathcal{D} = \{x^{(j)}, y_c^{(j)}, y_r^{(j)}\}_{j=1}^N. \qquad (4)$$

Then we maximize the a preference policy as follows:

$$\max_{\theta'} \mathbb{E}_{x,y} \left\{ r_\phi(x, y) - \beta \mathbb{D}_{\text{KL}}[\pi_{\theta'}(y|x)|\pi_{\text{ref}}(y|x)] \right\}. \qquad (5)$$

The reference model $\pi_{\text{ref}}$ are initialized by $\pi_{\text{SFT}}$, which prevents the learned parameter $\theta'$ from much deviation. The finally learned policy $\pi_{\theta'}(\cdot)$ are better equipped with human or user intentions.

DPO uses a closed form [33] derived from Eq. 5 to represent the optimal reward $r^*$ model by the learned optimal preference model $\pi^*$ as (with $K$ the constant factor):

$$r^*(x, y) = \beta \log \frac{\pi^*(y|x)}{\pi_{\text{ref}}(y|x)} + K. \qquad (6)$$

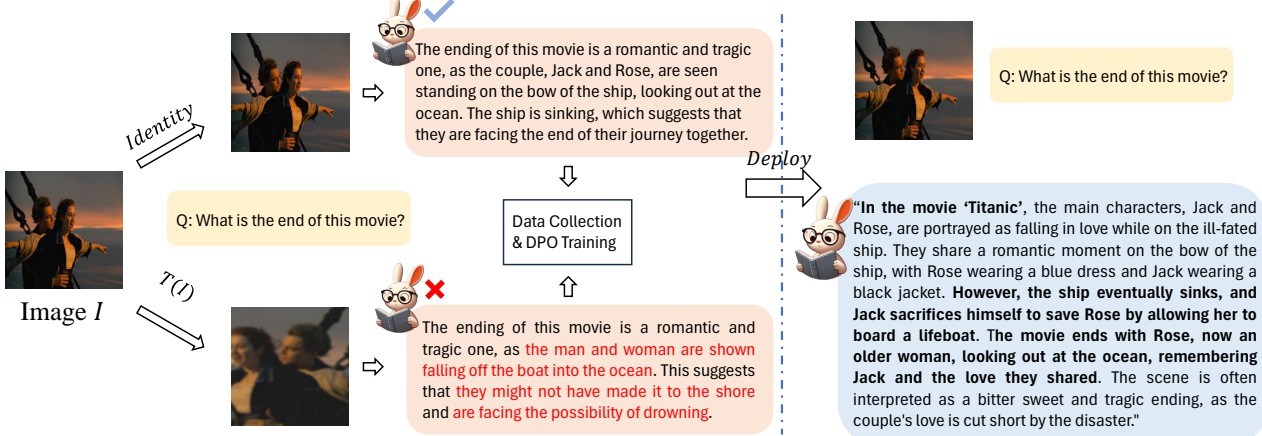

**Figure 3: The pipeline of SeVa. For each image $I$ in the selected dataset, we transform it with data augmentation $T$ to obtain the distorted one, while keeping a copy of the original image to form a pair. The shared questions are acted on the paired images to get the chosen and rejected responses, respectively, which undergo a data collection (e.g., filtering) process before the DPO training. In the left part, incorrect words or sentences are red color coded, while in the right part (the improved version of the model), we highlight excellent content with bold phase. Note that in the picture, we show the *same* image both for training and for testing, but actually the data distribution between them are different (cf. Sec. 4.1). This figure is best viewed in color.**

By substituting the $r^*$ in Eq. 3 with Eq. 6, we get the final optimized loss function of DPO as follows (cf. [33]):

$$\mathcal{L}_{\mathrm{d}} = -\mathbb{E}_{\mathcal{D}}\left[\log \sigma \left(\beta \log \frac{\pi_{\theta'}(y_c|x)}{\pi_{\mathrm{ref}}(y_c|x)} - \beta \log \frac{\pi_{\theta'}(y_r|x)}{\pi_{\mathrm{ref}}(y_r|x)}\right)\right] \quad (7)$$

The trainable parameter Eq. 7 is $\theta'$, which is the same with RLHF. But it do not need any online optimization process like PPO [34]. Instead, DPO samples $(x, y_c, y_r)$ from preference database $\mathcal{D}$ and optimize the VLM's parameters $\theta'$ with a simple classification loss.

## 3.2 Visual preference alignment

This section will introduce our pipeline: self-supervised visual preference alignment (SeVa). For a language model $\pi_{\mathrm{SFT}}$ after supervised finetuning stage, we feed it with the original and augmented image input as (for simplicity, we omit the generated token as conditional input shown in Eq. 2):

$$y_c^{(j)} = \pi_{\mathrm{SFT}}(g(I^j), q^j), \quad (8)$$

$$y_r^{(j)} = \pi_{\mathrm{SFT}}(g(\mathcal{T}(I^j)), q^j). \quad (9)$$

The $y_c^{(j)}$ and $y_r^{(j)}$ are regarded as a pair of raw chosen and rejected response. For each image $I^j$ and its question $q^j$, we generate only one paired preference data $(y_c^{(j)}, y_r^{(j)})$. Suppose there are total $M$ image-question pairs and its corresponding generated preference pairs, we simply filter those equaled responses to get remained preference data pairs (supposed $N_d$), which constitute the final preference database $\mathcal{D}_{\mathrm{SeVa}}$ for DPO training (cf. Alg. 1 and Eq. 7).

$$\mathcal{D}_{\mathrm{SeVa}} = \{I^j, q^j, y_c^{(j)}, y_r^{(j)}\}_{j=1}^{N_d}, \quad (10)$$

$$\text{s.t. } \forall j, \, y_r^{(j)}! = y_r^{(j)}. \quad (11)$$

We choose 6 different data augmentation strategies to construct our database and conduct DPO training. As shown in Table 1, all

self-constructed preference data are helpful for multi-modal comprehension, showing the validity of this pipeline. What surprises us is that training samples built from 'RandFlip' are quite helpful (e.g., an increase of 3.2% score on MMVet). We conjecture that this might derive from the natural property of the text data we used, since flipped images can make OCR and recognition harder and induce potential negative responses (cf. appendix for more details). Besides, all models have seen a noticeable improvement in adversarial settings of POPE, indicating the models trained by our self-supervised pipeline might be less inferior to potential interruptions.

## 3.3 Connection with contrastive learning

In contrastive learning, augmented views of the same image will share similar semantic in its deep embeddings, which optimize the following InfoNCE loss:

$$\mathcal{L}_{\mathrm{in}} = -\log \frac{\exp\left(q \cdot k_+/\tau\right)}{\exp\left(q \cdot k_+/\tau\right) + \sum_i^n \exp\left(q \cdot k_-^i/\tau\right)}, \quad (12)$$

in which $q$, $k_+$ are the positive embeddings from augmented views of the same image, while $k_-$ could be negatives embeddings from other images [5] or memory bank [13]. If we consider only one negative, and denote $f(q, k) = (q \cdot k)/\tau$ as the scoring function, we can reformulate Eq. 12 as:

$$\mathcal{L}_{\mathrm{in}}' = -\log \frac{\exp\left(f(q, k_+)\right)}{\exp\left(f(q, k_+)\right) + \exp\left(f(q, k_-)\right)}. \quad (13)$$

It is not hard to find that the optimzed loss function derived from Eq. 3 are quite similar with Eq. 13. The main difference lies in the *definition of negative samples*. Unlike contrastive learning, the predicted tokens from the augmented views of the same image will be counted as negatives, but not positives. If we imitate traditional InfoNCE loss in Eq. 12, we could inject multiple negative reward terms in Eq. 3 (by augmenting multiple views of the same image to

**Table 2: Comparison with state-of-the-art methods on 9 benchmark datasets. SeVa consistently improves LLaVA-1.5-7B/13B on 8 out of 9 benchmarks, namely, MMVet [40], LLaVA$^W$ [27], MMBench [28], MMBench-Chinese [28], POPE [21], SEED-Image [18], SHR [44], SQA [15], GQA [15]. Among them, SHR are newly proposed [44] to evaluate the hallucination ratio of VLMs.**

| Method | Language model | MMVet | LLaVA$^W$ | MMB | MMB$^{CN}$ | POPE | SEED$^I$ | SHR ($\downarrow$) | SQA | GQA |
|---|---|---|---|---|---|---|---|---|---|---|
| BLIP-2 | FLAN-T5 | 22.4 | 38.1 | – | – | 85.3 | 46.4 | – | 61.0 | 41.0 |
| InstructBLIP | Vicuna-7B | 26.2 | 60.9 | 36.0 | 23.7 | – | 53.4 | - | 60.5 | |
| InstructBLIP | Vicuna-13B | 25.6 | 58.2 | – | – | 78.9 | – | 51.2 | 63.1 | 49.5 |
| Shikra | Vicuna-13B | – | – | 58.8 | – | – | – | – | – | – |
| IDEFICS-9B | LLaMA-7B | – | – | 48.2 | 25.2 | – | – | – | – | 38.4 |
| IDEFICS-80B | LLaMA-65B | – | – | 54.5 | 38.1 | – | – | – | – | 45.2 |
| Qwen-VL | Quen-7B | – | – | 38.2 | 7.4 | – | 56.3 | – | 67.1 | 59.3 |
| Qwen-VL-chat | Quen-7B | – | – | 60.6 | 56.7 | – | 58.2 | – | 68.2 | 57.5 |
| LLaVA | Vicuna-7B | 26.7 | 63.0 | 34.1 | 14.1 | – | 25.5 | – | 38.5 | – |
| LLaVA-1.5 | Vicuna-7B | 30.5 | 63.4 | 64.3 | 58.3 | 85.9 | 65.7 | 36.7 | 66.8 | **62.0** |
| SeVa-7B | Vicuna-7B | **37.2** | **72.2** | **65.6** | **59.2** | **86.7** | 65.8 | 34.9 | 67.5 | 60.7 |
| LLaVA-1.5 | Vicuna-13B | 35.4 | 70.7 | 67.7 | 63.6 | 85.9 | 68.2 | 37.2 | **71.6** | 63.3 |
| SeVa-13B | Vicuna-13B | **41.0** | **80.1** | **68.7** | **64.8** | **87.4** | **68.6** | 36.6 | 71.2 | **63.4** |

produce multiple responses), and thus derive a more general form of DPO. Please refer to appendix for more details, and we will leave this as future work.

## 4 EXPERIMENT

In this section, we conduct experiments to verify the effectiveness of the proposed SeVa pipeline. Firstly, we provide the experimental settings (e.g., the data construction process). Then we show the results on multi-modal comprehension benchmarks. Finally, we provide fruitful ablations and visualizations to analyze SeVa with more details. For simplicity, *we name LLaVA-1.5-7B/13B trained with SeVa pipeline as SeVa-7B, SeVa-13B, respectively.*

### 4.1 Settings

**Data construction.** The source data we obtained are from LLaVA-665k SFT dataset [26], and we choose image-question pairs from TextVQA and OCRVQA [30, 35] (denoted as 'text+ocr') in 665k to generate the DPO preference data. *This setting is kept by default throughout all the experiment unless otherwise noted.* For each data instance in 'text+ocr' of LLaVA665k, we randomly choose 2 questions in its multi-turn dialogue, and pair each question with the image. We treat each image-question pair as *one data instance*, which leads to nearly doubled instance as the original 'text+ocr'. Then we randomly choose 8k such image-question pairs in 'text' and 'ocr' each, to form the original data source of 16k. Finally, these 16k image-question instances will go through SeVa pipeline to produce preference data with filtering (cf. Alg. 1), which are cut down by half to about 8k. For clarity in our later ablations, we denote data nums as those unfiltered preference pairs instead of the filtered ones (e.g., 16k but not 8k), since the num of raw unfiltered data is fixed and immune to filtering process (cf. appendix).

**Data augmentation choice.** As verified in Table 1, all the selected data augmentations are all helpful for multi-modal comprehension. In our later experiment, we choose *diffusion noise* as the default augmentation in SeVa pipeline, since it can easily control the intensity of distorted level, which is helpful for ablations (cf.

Fig. 6). The noise steps for training SeVa-7B and SeVa-13B are set as 800 and 500, respectively. As MOCO augmentation strategy is also highly effective as shown in Table 1, we are glad to verify more data augmentation strategies in our future work.

**Training details.** Following previous works [4, 22, 23, 38], we choose LLaVA-1.5-7B/13B as our base models in all our experiments. The LLaVA's weights are pretrained and SFT tuned before our DPO training, which is the by default pipeline as in many previous literatures [12, 33, 34, 44]. We adopt deepspeed ZeRO stage-3 [26] during DPO training and use Vicuna-7B/13B, CLIP-VIT-L-336px as our LLM and vision encoder, respectively. The total epoch, batch size, learning rate and weight decay are set as 1, 128, 2e-6 and 0, respectively, following previous work [44]. The hyper-parameters of lora $r$ are set as 1024, and the scale parameter $\beta$ in DPO is fixed as 0.1. DPO optmization are conducted with 8-A800 GPU, with LLaVA-7B/13B tuned for about 0.7/1.3 hours.

### 4.2 Compare with state-of-the-art

Firstly, we compare the proposed SeVa with state-of-the art VLMs. A total of 9 benchmarks are included, with multi-modal benchmarks as: MMVet [40], LLaVA-Bench-in-the-wild [26] (LLava$^W$), MMBench [28] (MMB), MMBench-Chinese (MMB$^{CN}$), POPE [21], SEED-Bench-Image [18] (SEED$^I$) and SHR [44], and traditional Question-Answer benchmarks ScienceQA [29] (SQA) and GQA [15]. Among them, MMVet, LLaVA$^W$ are two GPT-4 evaluated benchmarks. MMB and MMB$^{CN}$ are multiple choice question answer dataset that requires minimal GPT-4 involvement (e.g., answer choice aligning [28]). POPE and SHR are two hallucination benchmarks, and SQA & GQA are two traditional QA benchmarks.

As shown in Table 2, SeVa consistently improve the LLaVA-1.5 7B/13B models on 8 out of 9 multi-modal benchmarks. Specifically, it improve its 7B/13B models by 6.7%/5.6% on the most complex multi-modal dataset MMVet, showing the great power of the proposed pipeline. The same can be observed in LLaVA$^W$, where SeVa-13B achieves a 80% relatively score to GPT-4, boosting LLaVA-1.5-13B by a large margin of 9.4%. The great boost on GPT-4 evaluated datasets

**Table 3: Comparison with continual ('cont.') SFT in a 7B setting. We randomly select 15k (2%), 66k (10%) and 132k (20%) out of LLaVA665k to apply SFT. 'Unsupervised' means no answering labels are required. We also sample *all* the TextVQA and OCRVQA data in LLaVA665k for comparison (since SeVa utilize its unsupervised data version). * means the unfiltered preference instances (the actually kept DPO data are mostly by half, cf. Sec. 4.1). Training cost are all evaluated in a same experiment settings.**

| Method | Unsupervised | data source | data scale | MMVet | MMB | POPE | Training cost |
|---|---|---|---|---|---|---|---|
| LLaVA-1.5 | — | — | — | 30.5 | 64.3 | 85.9 | — |
| cont. SFT (2%) | ✗ | LLaVA665k | 15k | 31.9 | 63.7 | 86.0 | 0.1h |
| cont. SFT (10%) | ✗ | LLaVA665k | 66k | 32.8 | 64.9 | 86.0 | 1.2h |
| cont. SFT (20%) | ✗ | LLaVA665k | 132k | 33.9 | 64.2 | 86.1 | 2.4h |
| cont. SFT | ✗ | text+ocr | 102k | 32.5 | 65.2 | 86.7 | 1.2h |
| SeVa (*ours*) | ✓ | text+ocr | 8k* | 34.8 | 65.3 | 86.2 | 0.3h |
| SeVa (*ours*) | ✓ | text+ocr | 16k* | **37.2** | **65.6** | **86.7** | 0.7h |

**Table 4: MMVet evaluation score with different data (image-question) pairs and scale to conduct the SeVa-7B pipeline. The highest score in each dataset are highlighted with bold symbol. In SeVa pipeline, we choose TextVQA+OCRVQA by default. All the data listed are sourced from LLaVA665k [26].**

| Unspervised data | Data Scale | | | |
|---|---|---|---|---|
| | 2k | 4k | 8k | 16k |
| TextVQA [35] | 31.8 | 32.1 | 34.8 | **35.8** |
| OCRVQA [30] | 32.1 | 32.3 | 32.8 | **34.5** |
| GQA [15] | 31.4 | 31.8 | 34.1 | **35.9** |
| COCO [25] | 31.7 | 31.2 | 34.2 | **34.4** |
| Visual Genome [16] | 31.4 | 32.0 | 33.4 | **34.1** |

suggested a better alignment with user-intentions, as demonstrated in Fig. 5 and Fig. 7. We also observe a consistent improvement on POPE and SHR, which indicates the effect of SeVa to relieve object hallucinations, and thus we *may not* resort to specialized hallucination methods [17, 44]. Please note that the improvement on hallucinations are *not marginal*, since previous work that constructed huge amount of SFT data can lead to *decreased performance* on POPE (e.g., LVIS-Instruct4V [38]). There is also a higher multiple choices accuracy in MMB, MMB$^{CN}$ and SEED$^I$, showing the broad adaptability of SeVa. In Table 2, a little bit drop was found on traditional benchmarks SQA/GQA. We conjecture that this phenomenon might arise from a trade-off between stronger instruction following ability in traditional QA and better comprehension ability in modern VLMs benchmarks, which is also pointed out in [12] that stronger instruction following do not guarantee a more helpful and better VLMs. In fact, how to better trade-off between benchmarks remains a discussing issue [4] and is out-of-scope of this paper.

### 4.3 Ablations

Now we ablate the factors in our pipeline, to help readers better understand the success behind SeVa and its generalized ability. We choose SeVa-7B as the default model unless otherwise noted.

**Compare with SFT.** We start from comparing with continual SFT, as this is the most straight forward method to verify its effectiveness with supervised training (note SeVa *do not* need target answer labels). We conduct three different types of SFT: a random

**Representative questions in each dataset:**

**Question textvqa:**
- Provide a one-sentence caption for the provided image. Reference OCR token: [task-specific prompt]

**Question ocrvqa:**
- Who is the author of this book?
- What is the title of this book?

**Question gqa:**
- Is the cheese to the left of the food on the plate?
- Which kind of furniture are the cups hanging on?

**Question coco:**
- Where is the cat positioned in the image?
- Please provide a short description for this region: [task-specific prompt]

**Question vg:**
- Please provide a short description for this region: [task-specific prompt]
- Provide the bounding box coordinate of the region this sentence describes. [task-specific prompt]

**Figure 4: Illustration of representative questions in five databases from LLaVA665k. In our main experiment, we adopt a combination of 'textvqa' and 'ocrvqa'. The results of applying the other 3 question types can be found in Table 4.**

selection of 2%/10%/20% in LLaVA665k and a collection of TextVQA and OCRVQA in 665k (denoted as 'text+ocr'). We evaluate its downstream performance on MMVet, MMB and POPE, which can be found in Table 3. For a fair comparison, we strictly follow LLaVA-1.5's SFT settings during experiments. It can be observed that all SFT data are helpful for multi-modal comprehension. Specifically, text related data are more helpful for dealing with model hallucinations (e.g., on POPE). In comparison, our SeVa achieves the overall best results on all benchmarks with both less training time cost and data usage. More importantly, SeVa enjoys its unsupervised nature, making it more generalizable in reality.

**Generalize to other datasets.** Since we adopt the questions in TextVQA and OCRVQA as a default fashion, we now ablate on more dataset selection on top of these two dataset. For more

**Table 5: GPT-4 evaluated consistency score from 1-10. Q-consistency measures how the model's answers are aligned with the asked question, while A-consistency measures how answers ($t > 0$) are aligned with its generated tokens without sampling ($t = 0$). We adopt 1.5 version of LLaVA.**

| temp ($t$) | Q-Consistency (↑) | | A-Consistency (↑) | |
|---|---|---|---|---|
| | LLaVA | SeVa | LLaVA | SeVa |
| 0.2 | 7.30 | **7.75** | 6.45 | **6.95** |
| 0.4 | 7.11 | **7.98** | 6.28 | **7.32** |
| 0.5 | 7.39 | **7.93** | 6.52 | **7.29** |
| 0.7 | 7.81 | **8.18** | 7.10 | **7.64** |
| 0.9 | 8.43 | **8.53** | 7.87 | **7.98** |
| 50 | 7.75 | **8.47** | 6.94 | **8.03** |

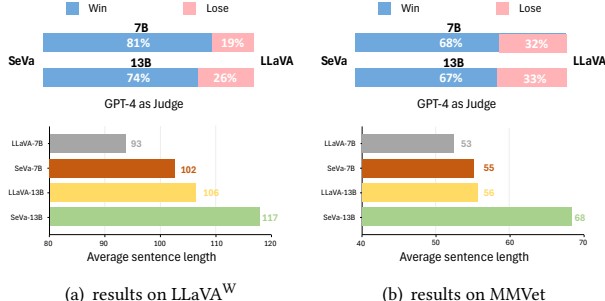

(a) results on LLaVA$^W$      (b) results on MMVet

**Figure 5: The pair-wise competition and output sentence length (by token) in LLaVA$^W$ and MMVet. We compare between SeVa and LLaVA-1.5 models in 7B and 13B settings.**

clarity, we choose other part of LLaVA665k including GQA [15], COCO [25] and Visual Genome [16] to form image-question pairs. We randomly select 2k-16k these data and conduct SeVa-7B training pipeline. The results are then evaluated on MMVet. As shown in Table 4, all the choosen dataset lead to consistent improvement over LLaVA-1.5-7B (30.5% score), showing the generalized ability of SeVa to different image and question domains. Besides, involving more unsupervised data leads to consistent better performance, which indicates the great power of SeVa towards further scaling.

**Why SeVa works?** Attentive readers might have a natural question: *why rejecting inferior answers lead to better model output than the original?* Here, we conduct a detailed experiment to show that, SeVa can be viewed as a special form of model calibration and rejecting negatives *does help*. Specifically, we collect image-question pairs from LLaVA$^W$ to prompt LLaVA and SeVa with higher sampling temperature $t$. Note that $t=0$ means no sampling in token generation, which is the default and most stable evaluation settings. Then we utilize GPT-4 (cf. appendix for prompt) to evaluate the its output consistency with scoring from 1 to 10, namely Q-consistency (how the model's answer are aligned with the input questions) and A-consistency (how the model's answer are aligned with its generated tokens w/o sampling). As shown in Table 5, the consistency score of both models fluctuates as $t$ grows. However, SeVa has seen an

**Table 6: The effect of hyper-parameter in LoRA [14] during DPO training. We range the rank of $r$ from 64 to 2048, while keeping the weight ratio fixed as 2 ($\alpha$ is always twice as $r$).**

| LoRA $r$ | LoRA $\alpha$ | MMVet | MMB | POPE |
|---|---|---|---|---|
| — | — | 30.7 | 64.3 | 86.8 |
| 64 | 128 | 32.8 | 64.7 | 86.5 |
| 128 | 256 | 33.9 | 65.0 | 86.4 |
| 256 | 512 | 34.0 | 65.1 | 86.6 |
| 512 | 1024 | 35.5 | 65.5 | **86.8** |
| 1024 | 2048 | **37.2** | **65.6** | 86.7 |
| 2048 | 4096 | 33.5 | 65.0 | 84.5 |

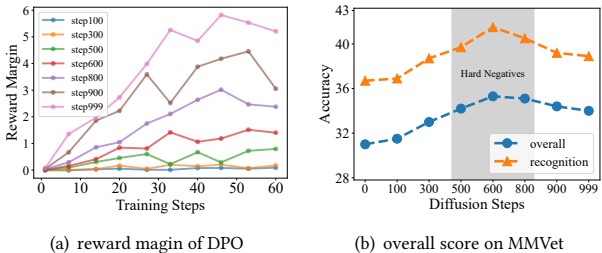

(a) reward magin of DPO      (b) overall score on MMVet

**Figure 6: Exploration of hard-negatives. We plot DPO training margin between chosen and rejected samples generated from different noise steps (cf. 6(a)), and evaluate their model's MMVet score after DPO training (cf. 6(b)).**

consistent better score in all different $t$s, especially when temperature becomes higher. Since VLMs are easily affected by randomness in generating tokens [20], SeVa has potentially made a calibrating effect by reducing the sampled probability of those *bad* tokens, thus being more tolerable to interruption. As a result, the model trained with SeVa can produce more robust and correct answers.

Besides, we also observed a longer response in LLaVA$^W$ and MMVet (cf. Fig. 5). This indicates that models trained with SeVa pipeline could potentially produce more detailed and meaningful responses that greatly benefits multi-modal comprehension.

**Hard negatives matter.** As we previously observed the phenomenon of hard negatives in Table 1, we now design a controllable experiment to study how strength of data augmentation affect the final training performance. Specifically, diffusion gaussian noise with different steps are chosen to generate preference data for DPO training, which are then evaluated on MMVet benchmark. As shown in Fig. 6(a), larger noise steps will increase the reward margin (cf. DPO [33] for more details) during training, indicating the increased preference divergence. In the meanwhile, a trade-off exists in the noise steps (possibly between 500 and 800), as shown in Fig. 6(b). We thus attribute the preference samples produced within this regime as *hard negatives* that are most valuable for DPO training.

**Effect of Lora.** For completeness, the hyper-parameters of LoRA are ablated on three multi-modal comprehension benchmarks, as shown in Table 6. In PETL [9], this parameter can be quite important, as it decides how much new knowledge the model can be absorbed during finetuning. As shown in the table, a relatively

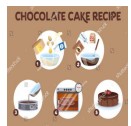

**Q: How many gallons of supreme gasoline can I get with $50?**

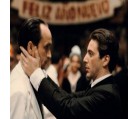

**Q: Should I add sugar when mixing egg and milk?**

**Q: Can you give a short introduction to this movie?**



**Q: What are the meals that I can cook with these?**

**LLaVA-1.5-13B:** Based on the image, you can get 3.6 gallons of supreme gasoline with $50. The price per gallon is $1.36, and $50 is enough to purchase 3.6 gallons.

**LLaVA-1.5-13B:** Yes, you should add sugar when mixing egg and milk in the chocolate cake recipe. This is because sugar helps to balance the flavors and create a more harmonious taste in the final product.

**LLaVA-1.5-13B:** The image features two men in suits, standing close to each other and looking at each other intently. They appear to be having a conversation or sharing a moment of connection. The scene, suggests that it might be a social gathering or a formal event. The men's attire and the presence of other people in the room indicate that this could be a scene from a movie or a special occasion.

**LLaVA-1.5-13B:** With these ingredients available in the refrigerator, you can prepare a variety of meals. The refrigerator contains several carrots, which can be used as a healthy snack or incorporated into dishes like salads, stir-fries, or soups.

**SeVa-13B:** To determine how many gallons of supreme gasoline you can get with $50, we need to know the current price per gallon at the gas station. *In the image, the price of supreme gasoline is $3.65 per gallon. With $50, you can buy:*

*$50 / $3.65 per gallon = 13.7* gallons of supreme gasoline

So, you can get approximately 13.7 gallons of supreme gasoline with $50.

**SeVa-13B:** *No, you should not add sugar when mixing egg and milk in the chocolate cake recipe. The sugar is typically added later in the process*, during the mixing of the dry ingredients, such as flour, cocoa powder, and baking powder. Mixing sugar with egg and milk at this stage can result in a runny, unstable batter that may not hold its shape well during baking. *It is essential to follow the recipe steps in the correct order* to achieve the desired texture and consistency of the cake batter.

**SeVa-13B:** *The image depicts a scene from the movie "Casablanca," featuring two main characters, played by Humphrey Bogart and Sidney Greenstreet, in a tender moment. Bogart's character, Rick Blaine, has his hand on Greenstreet's character, Victor Laszlo's,* shoulder, with their faces close together. This scene captures the emotional connection between the two characters, as they share a moment of understanding and camaraderie amidst *the turmoil of the World War II era in Casablanca.*

**SeVa-13B:** With the variety of food items in the refrigerator, you can prepare a range of meals and snacks, including:
*1. Fruit salad:* Combine the strawberries and blueberries to create a refreshing fruit salad ...
*2. Yogurt parfait:* Use the yogurt as a base for a delicious yogurt parfait, Layer the yogurt with granola, fresh fruit,...
*3. Carrot sticks with dip,* Slice the carrots into sticks and serve them with a healthy dip, like hummus, guacamole, or a yogurt-based dip.
*4. Smoothie:* Blend the yogurt, fruit, and any additional ingredients like nuts, seeds, or protein powder to create a nutritious and delicious smoothie.

Ability: OCR, Math, COT          Ability: Hallucinations (↓), OCR          Ability: Knowledge, Instruction Following          Ability: Helpfulness, Details

**Figure 7: Four examples to illustrate the improved ability of our SeVa, including chain of thought (COT), stronger OCR, less hallucinations, world knowledge and more detailed and clearer responses. Since we built-on LLaVA-1.5, we directly make comparison between them to show the effectiveness of our proposed method. For clarity, we highlight the incorrect response in LLaVA with red color, and emphasize ours with blue italic character. This figure is best viewed in color.**

higher $r$ is helpful to downstream tasks, showing the increased acquired knowledge of the VLM. However, too large $r$ will decrease performance, which can be attributed to the catastrophic forgetting in training a LLM (a similar point also pointed out in [4]). In conclusion, we set $r$ as 1024 to seek a best trade-off, and always keep the delta importance as 2 (the $\alpha$ is always twice as $r$).

**Aligning with user-intentions.** Finally, we visualize the improved VLM with SeVa. We selected image-question pairs from MMVet and LLaVA$^W$, which are sent to the original VLM (LLaVA-1.5-7B) and the improved model (SeVa-7B), respectively. As shown in Fig. 7, Our SeVa demonstrates superiority over LLaVA in various aspects: stronger OCR ability, where SeVa could recognize the exact number of the gas price; less hallucinations can be observed in the second picture, where SeVa accurately comprehends the process of 'chocolate cake recipe' and produces correct answers. It is also surprising that SeVa could potentially recovers more world knowledge after DPO training, as it gives detailed and accurately response to the introduction of a movie (the third picture). Finally, we found that through our DPO training, the models provide more detailed and helpful answers, as suggested by the last picture of a meal ordering. Following Vicuna [6], we also conduct a competition game between SeVa and LLaVA-1.5 under 7B and 13B settings on LLaVA$^W$ and MMVet, respectively. We introduce GPT-4 as judge to evaluate the score of each model's responses (similar to the evaluation process in these two benchmarks) and calculate the statistics of 'win' and 'lose'. *This could serve as a result that indicates how the*

*models are aligned with user-intentions* [45]. Note we didn't include 'tie' [6], since we observed a majority of meaningless 'tie', where both SeVa and LLaVA achieves a a score of *zero*. As shown in Fig. 5, SeVa has occupied most of the winning rounds in comparison with LLaVA, across both model sizes and benchmark datasets. Together with the visualizations in Fig. 7, we believe that our SeVa could probably serve as a strong pipeline to improve current VLMs' chat ability, and makes them more suitable towards practical usage.

## 5 CONCLUSIONS AND FUTURE WORK

In this paper, we introduce SeVa: Self-supervised visual preference alignment, for multi-modal comprehension. It first went through a data collection and filtering process with response from the original and distorted image, respectively. Then, the standard DPO training are applied to improve the model's capability. Experiments in various benchmarks clear verify and efficiency of the approach. We also conduct ablations and quantitative visualizations to reveal the latent mechanism of SeVa, which demonstrate the great merit of aligning large models in vision-language domains.

In the future, we might consider generalize SeVa to various domains. We will study the effect of data scale: how the model will benefit from DPO training if more unlabeled data are involved, On top of that, we would investigate the potential to generalize our SeVa pipeline RLHF [31] domain. (e.g., whether the generated preference data could be helpful in building a reward model).

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
