# OpenReview forum: "Self-Supervised Visual Preference Alignment"
_acmmm.org/ACMMM/2024/Conference — MM2024 Oral_

### Official Review · Reviewer_XTxx · 2024-05-22

**Rating:** 5
**Confidence:** 3

**Summary:**

This paper introduces a data augmentation method specifically designed for self-supervised learning, which is particularly aimed at large language models (LLMs). They propose a method that employs the strategy of rejecting generated samples, which are the same answer with the chosen data, to enhance the quality of augmentation. This approach achieves significant improvement on MMVet and $LLaVA^{W}$. The ablation studies show that the generation, consistency score, hyper-parameter in LoRA, and hard-negatives.

**Strengths:**

1) This method is easily understand and reproduce.
2) The self-supervised method can be adapted to other multi-modal areas.
3) The approach leverages a few samples to make a great improvement on same datasets. (e.g., 16k vs. 102k on POPE).
4) The authors conduct substantial compared experiments on different metric and with different setting, and the results are encouraging.

**Limitations:**

1) This approach is only conducted on LLaVA, which cannot demonstrates its generation on different models.
2) It is appreciate for the comparison experiments about data scale (8k and 16k in Table 3, and generation in Table 4). Although the motivation is to maximize performance with minimal data usage, the upper bond performance of this method is unclear for readers on the data scale metric.

Typo errors:
The quotation mark in the caption of Figure 2.

**Suitability:**

3

---

### Official Review · Reviewer_fxSP · 2024-05-22

**Rating:** 3
**Confidence:** 4

**Summary:**

This paper presents a novel self-supervised approach to visual preference alignment, which aims to address the preference alignment problem in Visual-Language Models (VLMs) in an unsupervised context. Researchers generate chosen and rejected responses based on the original and augmented image pairs, respectively, and carry out preference alignment employing direct preference optimization (DPO). The underlying principle involves the use of carefully designed image augmentations that lead VLMs to generate false yet challenging negative responses. This, in turn, facilitates the model's learning process, enabling it to produce more robust and potent answers. The entire procedure is no longer dependent on supervision from GPT-4 or human involvement during the alignment phase, and it is highly efficient, requiring only a few lines of code to execute effectively.

**Strengths:**

1. This work marks the first attempt at unsupervised preference alignment within VLMs, breaking free from reliance on either GPT-4 or expensive human-annotated data, charting a new path for preference alignment.

2. The proposed SeVa methodology is achievable with just a few lines of code, and introduces an efficient data construction pipeline.

3. Through abundant experimental data and illustrative visualization results, the paper extensively demonstrates SeVa's capability to enhance VLMs across various dimensions.

**Limitations:**

This paper innovatively presents a self-supervised method for visual preference alignment and validates its efficacy across several pivotal benchmarks. Nevertheless, there remains room for further refinement and deeper investigation, especially in terms of experimental details, selection of data augmentation strategies, choice of evaluation metrics, and theoretical elucidation.

1. The details of the experimental setup are not sufficiently clear, including the criteria for selecting specific image augmentation strategies and the precise implementation details of DPO. Moreover, the model diagram is too simple and should show the model network structure and the pipeline of data processing in the model.

2. The research does not offer a systematic approach to selecting the most appropriate augmentation strategies for different tasks or datasets. This may lead to increased practical difficulty in finding optimal augmentation when applied in new environments.

3. The evaluation of GPT-4 itself may also introduce some subjectivity, especially for complex reasoning tasks. Additionally, the experiments may overlook neutral responses where the model fails to distinctly differentiate between superior and inferior options, which may affect the comprehensiveness of model performance evaluations.

4. Although the authors propose SeVa as a special form of single-negative sample contrastive learning, the theoretical exploration remains superficial with no rigorous proof, thereby limiting a deep understanding of its working principles.

**Suitability:**

3

---

### Official Review · Reviewer_MeA6 · 2024-05-24

**Rating:** 6
**Confidence:** 3

**Summary:**

The paper introduces a method that optimizes vision-language models by self-generated chosen or rejection pairs with DPO. It designs augmentation to the image to generate false but hard negative answers. The method does not rely on human annotation for alignment. Extensive experiments and analyses show the proposed method is effective.

**Strengths:**

- The paper is well-written and easy to follow.
- The proposed methods are novel and effective.
- The analyses of the connections to contrastive learning are insightful.
- The methods proposed do not rely on human or GPT-4 annotations, which is a more elegant and scalable approach.
- The paper conducts extensive analysis, such as comparisons with SFT and the importance of hard negatives, which helps to better understand why SeVa is effective.

**Limitations:**

- I've noticed a phenomenon where after DPO, the outputs of the model seem to have become longer. Could this affect the evaluation metrics, especially those that rely on LLM evaluation, as LLM might prefer longer answers?
- The augmentation for creating negative examples seems to be degradation operations from the CV field. Could adding more purposeful and interpretable operations, such as replacing objects or changing attributes,  potentially enhance the performance, as this would enhance the model's accurate understanding of visual content?

**Suitability:**

3

---

### Meta-Review · Area_Chair_NgG6 · 2024-07-02

**Recommendation:** Accept (Oral)
**Confidence:** 5

**Metareview:**

This paper investigated unsupervised preference alignment in Vision-Language Models (VLMs). The idea is simple but intuitive: properly designed image augmentation will induce VLMs to generate hard negative responses. Motivated by this idea, this work further proposed self-supervised visual preference alignment that takes the original and augmented images for both supervised fine-tuning and direct preference optimization. Experiments on 9 benchmark datasets show the significant improvement of the proposed SeVa compared to LLaVA-1.5 on 7B and 13B models.

This paper received 1 Accept, 1 Weak Accept, and 1 Borderline Reject. After rebuttal, the reviewer who gave borderline reject increased the score to borderline accept. Considering the strengths on motivation, novelty, effectiveness, and writing, the final recommendation is a clear accept.